# Molecular Biomarkers in Ocular Graft-versus-Host Disease: A Systematic Review

**DOI:** 10.3390/biom14010102

**Published:** 2024-01-12

**Authors:** Jerry Bohlen, Charlyn Gomez, Jason Zhou, Fernando Martinez Guasch, Caitlyn Wandvik, Sarah Brem Sunshine

**Affiliations:** Department of Ophthalmology and Visual Sciences, University of Maryland School of Medicine, Baltimore, MD 21201, USA; jbohlen@som.umaryland.edu (J.B.); charlyn.gomez@som.umaryland.edu (C.G.); jwzhou@som.umaryland.edu (J.Z.); fmartinezguasch@som.umaryland.edu (F.M.G.); caitlyn.wandvik@hsc.utah.edu (C.W.)

**Keywords:** biomarkers, ocular graft-versus-host disease, cytokines, proteomics, dry eye disease

## Abstract

Ocular graft-versus-host disease (oGVHD) affects ~50% of post-stem cell transplant patients and is the only form of GVHD diagnosed without a biopsy. As it must be distinguished from other dry eye diseases, there is a need to identify oGVHD biomarkers to improve diagnosis and treatment. We conducted a systematic review of 19 scholarly articles published from 2018 to 2023 including articles focused on adult patients diagnosed with oGVHD following allogeneic hematopoietic stem cell transplant and used biomarkers as the outcome measure. Articles that were not original investigations or were not published in English were excluded. These clinical investigations explored different molecular oGVHD biomarkers and were identified on 3 October 2023 from the Scopus, PubMed, and Embase databases by using search terms including ocular graft-versus-host disease, biomarkers, cytokines, proteomics, genomics, immune response, imaging techniques, and dry-eye-related key terms. The Newcastle–Ottawa scale for case–control studies was used to assess bias. From the 19 articles included, cytokine, proteomic, lipid, and leukocyte profiles were studied in tear film, as well as ocular surface microbiota and fluorescein staining. Our findings suggest that cytokine profiling is the most studied oGVHD biomarker. Additionally, variations correlating these biomarkers with disease state may lead to a more targeted diagnosis and therapeutic approach. Limitations include language bias, publication bias, and sampling bias, as well as a lack of appropriate controls for included studies.

## 1. Introduction

A biomarker is defined as an objectively measurable parameter produced by a biological system that can indicate both physiological and pathological processes [1]. They can also manifest as a response to treatments or interventions. Thus, biomarkers represent a wide range of categories from biomolecules, such as proteins, to tissues or blood samples. Although it is critical to understand the pathogenesis of different diseases to improve clinical diagnostics, treatment, and outcomes, our knowledge of related biomarkers remains limited for certain pathologies. One such disease is the ocular manifestation of graft-versus-host disease (GVHD).

Cancer patients who undergo allogeneic hematopoietic stem cell transplantation (allo-HSCT) are at significant risk for developing GVHD, which involves donor T-cell attacks on the host’s healthy tissues. The chronic phase of this condition, known as chronic GVHD (cGVHD), can affect multiple organ systems, including the skin, lungs, liver, gastrointestinal tract, oral mucosa, and the eyes—which is ocular GVHD (oGVHD) [2].

The ophthalmic manifestations of this disease occur in more than half of all allo-HSCT patients, which may present as severe dry eye disease, foreign body sensation, corneal scarring, decreased visual acuity, and, in severe cases, corneal perforation [3]. In the clinic, these signs and symptoms are evaluated through several methods and scored according to the International Chronic Ocular GVHD Consensus Group (ICOGCG) diagnostic criteria outlined in Table 1 [3]. Patients report decreased quality of life due to pain and discomfort that is measured with validated symptomatic questionnaires such as the Ocular Surface Disease Index (OSDI) score. Assessments include measurement of tear production (Schirmer’s) and conjunctival injection, while corneal fluorescein staining identifies characteristic signs such as keratoconjunctivitis sicca (KCS), cicatricial conjunctivitis, or regions of punctate keratopathy. According to the National Institutes of Health (NIH), the current method for diagnosing oGvHD involves the presence of at least one identifiable clinical sign and an abnormal Schirmer test result [4].

In clinical settings, the limited specificity and consistency of these indicators highlight the challenges associated with diagnosing and characterizing oGVHD, which are necessary to optimize treatment and identify patients at risk for the development of the most severe forms of oGVHD. As such, validated biomarkers are a critical tool for objective diagnostics and monitoring tools for this disease.

Additionally, treatment options are limited, locally and systemically, to broad-acting topical steroids and nonspecific methods to improve symptoms and quality of the tear film. These interventions include the use of preservative-free artificial tears, serum tears, warm compresses, and environmental changes. Since many patients experience persistent oGVHD, it is crucial to better understand its biomarkers to predict which allo-HSCT patients are at increased risk of developing oGVHD, to differentiate oGVHD from other DEDs, and to assess oGVHD severity clinically.

Currently, the most common way to study biomarkers in oGVHD is through tear collection and analysis, though other methods include serum samples or imaging [5]. Tear samples are analyzed for cytokines, proteomics, lipid profiles, leukocytes, and microflora. This study systematically reviews the present understanding and use of biomarkers for risk assessment, diagnosis, and treatment of oGVHD.

## 2. Methods

### 2.1. Search Strategy

A systematic review, registered with Prospero (CRD42023488232), was conducted across three databases: PubMed, Scopus, and Embase. The search criteria encompassed terms related to oGVHD, biomarkers, immune responses, inflammation, and imaging techniques. Two common Boolean operators were created and applied in each database:(“ocular graft-versus-host disease” OR “ocular chronic graft-versus-host disease” OR “ocular cGVHD” OR “oGVHD” OR “ocular GVHD” OR “ocular graft versus host disease”) AND (biomark* OR cytokin* OR chemok* OR proteom* OR genom* OR transcriptom* OR metabolom* OR lipidom* OR “immune response” OR “immune profile” OR inflamm* OR “tear film” OR “ocular surface” OR “conjunctiva” OR “cornea” OR “lacrimal gland” OR “meibomian gland” OR “dry eye” OR “macula*” OR “fundus” OR “imaging modalit*” OR “optical coherence tomograph*” OR “OCT” OR “in vivo confocal microscop*” OR “corneal topograph*” OR “fluorescein angiograph*” OR “indocyanine green angiograph*” OR “fundus photograph*” OR “ultrasound biomicroscop*” OR “scanning laser ophthalmoscop*” OR “multimodal imag*” OR “imaging technique*”).(“chronic graft-versus-host disease” OR “acute graft-versus-host disease”) AND (“conjunctiva” OR “cornea” OR “lacrimal gland” OR “meibomian gland” OR “dry eye” OR “macula*” OR “fundus”) AND (biomark* OR cytokin* OR chemok* OR proteom* OR genom* OR transcriptom* OR metabolom* OR lipidom* OR “immune response*” OR “immune profile*” OR inflamm* OR “tear film” OR “ocular surface” OR “imaging modalit*” OR “optical coherence tomograph*” OR “OCT” OR “in vivo confocal microscop*” OR “corneal topograph*” OR “fluorescein angiograph*” OR “indocyanine green angiograph*” OR “fundus photograph*” OR “ultrasound biomicroscop*” OR “scanning laser ophthalmoscop*” OR “multimodal imag*” OR “imaging technique*”).

Abstracts were collected and filtered for the 2018–2023 timespan, English language publications, and full-text availability. This timeline was selected as the most up-to-date for developments in oGVHD research that have not been reviewed compared to earlier findings. Articles were extracted on 3 October 2023 and independently reviewed by two authors (J.B. and J.Z.) on 7 October 2023.

### 2.2. Selection Criteria

This review included studies meeting specific criteria: they were written in English, contained original clinical data published between January 2018 and September 2023, focused on adult patients diagnosed with oGVHD following allo-HSCT, and used biomarkers as outcome measures for oGVHD. Exclusion criteria comprised review articles, case reports, conference abstracts, editorials, letters, commentaries, or studies unavailable in full text. Additionally, studies involving participants without oGVHD, biomarkers unrelated to oGVHD, biomarkers not related to ocular manifestations, and biomarkers lacking defined outcome measures were excluded. Adherence to the Preferred Reporting Items for Systematic Review and Meta-Analyses (PRISMA) standards was maintained throughout the selection and review process.

### 2.3. Data Extraction and Quality Assessment

Each article was carefully reviewed by the authors to ensure they included biomarkers with well-defined outcome measures within the relevant clinical context. Limitations included language bias, selection bias, and publication bias.

Evidence quality assessment of the studies was conducted using the Newcastle–Ottawa Quality Assessment (NOQA). Two authors (J.B. and J.Z.) independently evaluated each article to determine the bias score on 9 October 2023. In cases of disagreement, a consensus was reached after a careful review of the wording and content within the primary literature.

## 3. Results

We conducted a comprehensive search across various databases, resulting in the identification of 307 relevant records. After the initial screening and removal of duplicates, 176 records remained for further evaluation. Following a detailed assessment of titles and abstracts, 36 studies met our initial inclusion criteria. Subsequent full-text evaluations resulted in 19 studies being included in the final review. The selection process is summarized in Figure 1.

After full-text assessment, 15 studies were excluded from this review due to the use of subjective measurements and no quantifiable biomarkers. Of note, five of the papers excluded for this reason looked at Meibomian gland architecture [6,7,8,9,10,11]. While a promising area of research, Meibomian gland dysfunction had no objective means for quantification and was not included in this review. The remaining excluded studies did not use reliable reproducible metrics in their investigation, used a human cell culture model instead of a clinical model [12], or only included one oGVHD subject [13].

We provide a detailed summary of each of the 19 studies included in Table 2. This table includes information on the biomarkers evaluated, sample types, patient groups, analysis techniques, comparison with control groups, correlation with clinical parameters, and corresponding references. Cytokines have emerged as the primary focus of investigation for biomarkers in oGVHD. Other prevalent biomarker categories include proteins, lipids, microflora, and leukocytes. Notably, more than half of the studies in our analysis collected tear samples to assess these biomarkers.

### Quality Assessment

The methodological quality of each study was assessed using the Newcastle–Ottawa case–control scale (Table 3). The eight criteria, each worth 1 point, include appropriate case definition, representativeness of the cases, selection of controls, definition of controls, comparability of cases and controls, ascertainment of cases, the same method of ascertainment for cases and controls, and non-response rate. In total, eight studies scored six points, seven studies scored five points, and four studies scored four points. All but three studies included non-oGVHD patient controls. However, when comparing case and control sections, we selected patient history of allo-HSCT as the key factor to control for, as this allows us to directly compare oGVHD biomolecules against patients who have undergone allo-HSCT without developing ocular disease. Only seven out of twenty studies earned that distinction. Overall, most studies defined and chose a representative case population well and consistently attained information from secure records. The full breakdown of the point distribution for all the studies is shown in Table 3.

## 4. Discussion

### 4.1. Tears as a Valuable Resource for Biomarker Analysis

Based on the identified literature, the collection and analysis of tears have emerged as the predominant method for potential biomarkers to detect and monitor oGVHD. This approach holds significant promise in the field of ophthalmology due to several key advantages. Tears are a non-invasive and easily accessible bodily fluid and provide a convenient source of analysis for a wide range of molecules including cytokines, proteins, and lipids. This minimizes patient discomfort and facilitates frequent sampling, which is crucial for long-term assessment of a dynamic condition such as oGVHD.

### 4.2. Cytokine Profile in Tears

Several cytokines have been implicated in the development of oGVHD. The key for these biomarkers is to identify cytokines with a diagnostic value that correlates with disease severity. Differentiating patients with oGVHD from those with dry eye disease (DED) is important and may be challenging due to similar disease manifestations compared to differentiation of oGVHD and non-oGVHD allo-HSCT patients [14]. These distinctions are important for the treatment of oGVHD; therefore, a pre-transplant ophthalmic examination is critical to identify patients with baseline DED.

Frequently observed cytokine levels in the tears of oGVHD patients include elevation of IL-6 and IL-8 as well as decreases in IL-7 and epidermal growth factor (EGF). In areas of inflammation, IL-6 functions to regulate immune responses while IL-8 (specifically) activates neutrophils. IL-7 induces T-cell maturation and homeostasis; thus, a reduced level is expected in oGVHD patients via thymic injury from HSCT. Likewise, decreased amounts of EGF, which increases cell growth, may indicate an inflammatory state susceptible to both GVHD and oGVHD. Additionally, numerous studies were identified showing the correlation between cytokines MMP-9, granulocyte-macrophage colony-stimulating factor (GM-CSF), B-cell Activating Factor (BAFF), and ICAM in the tears of patients with oGVHD. Hu et al. investigated tear cytokines of oGVHD and DED patients by microsphere-based immunoassays and found an elevation of ICAM-1, IL1β, IL-6, and IL-8 in addition to decreased IL-7 and EGF in oGVHD patients relative to DED patients [14]. Clinically, the Schirmers test, TBUT, corneal fluorescein staining, and OSDI were measured and correlated with cytokines. They found a negative correlation between the Schirmers test and ICAM-1, IL-8, IL-6, and IL-1β despite a positive correlation between those cytokines and CFS and OSDI [14]. Some of these cytokines were also studied by Qiu et al. [31], where oGVHD patients were compared to healthy controls rather than DED patients. The study further investigated the role of complement in the diagnosis and severity of disease [31]. In addition to confirming the presence of elevated cytokines in oGVHD by Hu et al. [14], Qiu et al. also found increased BAFF and GM-CSF compared to healthy controls [31]. BAFF is released by innate immune cells to stimulate B-cell activity, and GM-CSF promotes monocytes and macrophages. Together, these cytokines may help explain and identify the inflammatory state of oGVHD. In contrast, due to elevated IL-7 and EGF in healthy controls relative to oGVHD patients, Qiu et al. suggest that these cytokines could protect against oGVHD [31]. This latter finding requires further investigation and should be studied in comparison to DED patients.

A third study also noted an increased level of IL-6 and IL-8 in oGVHD patients compared to non-oGVHD allo-HSCT and healthy patients [25]. In full, Nair et al. observed interferon γ, IL 6, IL 8, IL 10, IL 12AP70, IL 17A, MMP 9, and VEGF to be increased not only in oGVHD patients compared to non-oGVHD allo-HSCT and controls but also in the non-oGVHD allo-HSCT group compared to controls. Additionally, their findings suggest that high levels of MMP 9 and VEGF tears in allo-HSCT patients may indicate oGVHD, which would be useful in diagnosing the disease [25]. Similarly, Liu et al. investigated several overlapping tear cytokines: IL-6, IL-10, TNF-a, EGF, MMP-2, MMP-3, MMP-7, BAFF, and a proliferation-inducing ligand (APRIL). Their results in oGVHD and non-oGVHD allo-HSCT controls showed that EGF levels correlate with corneal epithelial changes observed in oGVHD, which had increased surface defects visualized with fluorescein staining [18]. While tears are popular to sample due to their non-invasive collection, it may also be useful to have systemic monitoring, such as via serum cytokines, of GVHD in oGVHD patients; Pietraszkiewicz et al. investigated cytokines in both tears and serum. They compared three groups: (1) oGVHD as defined by the 2014 NIH consensus criteria, (2) non-oGVHD post allo-HSCT, and (3) healthy controls. MMP-9 and CXCL10 were found to be elevated in the tears and serum of patients with oGVHD, while TNFa, IL-10, and MIP-1a were elevated and serum and IL-8 were increased only in the tears of patients with oGVHD [15]. Although the results were not statistically significant, Pietraszkiewicz et al. [15] demonstrated the importance of studying both the local and systemic cytokine profiles in the tears and serum due to differences found between them, suggesting varying forms of inflammation which may call for different therapeutic interventions.

There is emerging evidence for new biomarkers that have been sparely investigated, such as lymphotoxin-α (LT-α), Oncostatin M (OSM), and LIGHT/TNFSF14, that may hold valuable diagnostic and prognostic capabilities. Ma et al. sought to determine whether tear lymphotoxin-α (LT-α; formerly TNF-Beta), a pro-inflammatory cytokine, was correlated with oGVHD due to its expression only in cells implicated in systemic GVHD pathogenesis. After analyzing samples from oGVHD and healthy patients, it was reported that those with oGVHD had significantly lower LT-α than healthy individuals [29]. Regarding clinical assessments, LT-α negatively correlated with OSDI and positively correlated with TBUT [29]. Interestingly, this work highlights the need for studying cytokine pathophysiology more in depth to better understand their diagnostic and clinical value in oGVHD.

Shen et al. [30] also discussed 29 tear cytokine biomarkers for oGVHD. Shen et al. [30] compared the tear cytokine profiles of oGVHD and dry eye disease (DED) patients, finding significant differences in IL-2, IL-6, and IL-8 with a strong correlation to severity and ocular surface symptoms in the oGVHD patient population. B-cell Activating Factor (BAFF) correlates with oGVHD disease severity, while a proliferation-inducing ligand (APRIL) is inversely correlated with both BAFF and oGVHD disease severity. MMP3, ICAM-1, CD62E (E-selectin), and neuropilin-1 are also potential diagnostic biomarkers for oGVHD, as they correlate with OSDI scores and NIH grade. Further research and clinical validation are needed to confirm their utility and roles in oGVHD pathophysiology.

The study conducted by An et al. has identified several cytokines that could serve as potential biomarkers: Oncostatin M (OSM), neutrophil gelatinase-associated lipocalin (NGAL), and LIGHT/TNFSF14. These are all secreted by neutrophils during NETosis. Additionally, according to their report, greater disease severity is observed in oGVHD patients with an excess of neutrophils relative to epithelial cells in ocular surface washings [16]. OSM was found to be elevated in ocular surface washings of oGVHD patients compared to healthy participants. Their data suggest that, via OSM, NETs contribute to corneal barrier disruption, leading to accelerated cell death, desquamation, and exposure/sensitization of epithelial nociceptors. Additionally, this exposes the cornea to environmental factors, including pathogens and allergens, potentially contributing to oGVHD’s chronic inflammatory state. Significantly higher levels of NGAL have also been found in oGVHD patients as compared to non-oGVHD patients and healthy subjects. The authors report that this protein has been linked to meibomian gland atrophy, a significant clinical finding in oGVHD patients. Additionally, NGAL may contribute to eyelid margin pathology, characterized by keratinization, squamous metaplasia, and lichenoid changes, all observed in oGVHD patients.

LIGHT/TNFSF14 levels have also been detected in ocular surface washings of oGVHD patients and are a source of ocular surface inflammation. It is noteworthy that LIGHT/TNFSF14 has also been found in experimental NETs, suggesting that neutrophils may be the source of this cytokine during NETosis. LIGHT/TNFSF14 is known to cause T-cell activation and proliferation, making it a potential indicator of T-cell-driven immune activity in oGVHD [32,33]. These cytokines, when present in ocular surface washings and NETs, provide valuable insights into oGVHD’s disease mechanisms and severity. By further exploring these biomarkers, clinicians may enhance their ability to diagnose, monitor, and manage oGVHD effectively, ultimately improving the quality of care for patients affected by this condition.

### 4.3. Proteomic Profile in Tears

In addition to cytokines, other proteins have been studied in the tears of patients with oGVHD and may prove valuable in improving diagnosis and management. The most frequently observed decline of proteins in oGVHD tears compared to controls were lactotransferrin, lipocalin-1, lysozyme C, and proline-rich proteins 1 and 4. However, there were also various increased proteins in oGVHD, and the most frequently used method for tear protein analysis was liquid chromatography and mass spectroscopy. Gerber-Hollbach et al. [13] studied 79 proteins to evaluate the tear proteomics in patients with and without oGVHD who had undergone allo-HSCT. Nucleic acid binding and cytoskeletal proteins were highly upregulated while enzyme modulators and receptor proteins were downregulated [13]. Ciavarella et al. (2021) [17] analyzed the proteomic tear profile pre- and post-HSCT. Among the proteins investigated, they observed a significant change in total protein, lactoferrin, transferrin, and zinc-alpha-2-glycoprotein in the subjects who developed oGVHD [17]. O’Leary et al. studied post-HSCT patients and found that increased oGVHD severity was correlated with decreased lactotransferrin, lysozyme, polymeric immunoglobin receptor (PIGR), immunoglobin J chain, prolactin inducible protein, and immunoglobulin heavy constant alpha [24]. Clinically, decreased lysozyme and PIGR, which are defense proteins, were seen with high OSDI and corneal fluorescein staining scores [24]. Yet, there was also a positive correlation between these proteins and Schirmer and TBUT scores [24], which complicates the clinical utility of this finding.

Artificial intelligence (AI) can play a pivotal role in the study of oGVHD and potentially assist molecular biomarkers in diagnosing and monitoring oGVHD; O’Leary et al. [24] used machine learning to predict the severity of oGVHD. While these interpretations were challenging, they importantly identify machine learning as a potential tool to identify and optimize the clinical utility of biomarkers. Interestingly, O’Leary et al. [24] were the only investigators in our literature review who employed AI in their investigative approach. However, the algorithms of machine learning hold promise for processing and identifying oGVHD biomarker data sets, warranting further study.

### 4.4. Lipid Profile in Tears

Evidence has shown the critical role of lipids in tear film, ensuring tear film stability and preventing evaporation [34]. Therefore, investigating tear lipid profiles holds promise for understanding and diagnosing various ocular conditions, such as oGVHD, and may offer promising avenues for potential biomarkers. In Ma et al. [21] study, tear lipid metabolites were explored as potential diagnostic biomarkers for oGVHD using liquid chromatography–mass spectrometry. Significant metabolic variations were observed, particularly highlighting the dysregulation of glycerophospholipid metabolism, sphingolipid metabolism, and biosynthesis of unsaturated fatty acids. Key metabolites within these pathways, including PC (34:1), SM, LacCer, and DHA, emerge as potential biomarkers for oGVHD diagnosis and therapeutic targets.

The study emphasizes previous literature in great depth, explaining how glycerophospholipids and sphingolipids are essential components of biological membranes and important in the propagation of T-cell receptor (TCR) signaling [35] and appear to be associated with the allogeneic immune response in oGVHD patients. Their study also revealed elevated fatty acid synthesis in chronic oGVHD, stimulating immune cell proliferation and inflammatory cytokine production. This finding supports other research that identifies the importance of fatty acid synthesis in the development of GVHD in animal models [36].

The emerging evidence of tear lipid metabolites presents promising avenues for understanding, diagnosing, and managing oGVHD. Further exploration of these pathways may lead to novel diagnostic biomarkers and innovative therapeutic strategies, which hold the potential to provide more effective management to individuals with this debilitating condition.

### 4.5. Leukocyte Profile in Tears

OGVHD is known to be an inflammatory process, but quantification of the number of immune cells in the conjunctiva and tears was studied as a potential biomarker. Alba-Linero et al. [20] studied the presence of CD8+ lymphocytes in conjunctival tissue in patients with oGVHD. CD8+ lymphocytes were detected in conjunctival samples in both HSCT patients with and without GVHD but with a higher presence of CD8+ T-cells in patients with GVHD as well as increased CD8+ lymphocytes in patients with GVHD compared to healthy patients. Further research is needed to determine if CD8+ lymphocytes are an accurate and reproducible biomarker for oGVHD.

Localization of neutrophils in cGVHD patients may be a more specific biomarker for oGVHD. An et al. found that the tears of patients with oGVHD had a significantly increased number of neutrophils compared to healthy controls [16]. The patients who had more neutrophils than epithelial cells in their tears had more severe clinical symptoms of oGVHD as measured by OSDI, bulbar redness, and severity score. Neutrophils may be a valuable biomarker to quantify, monitor disease severity, and understand localization of tissue damage on the ocular surface.

### 4.6. Microflora Profile on the Ocular Surface

Emerging evidence supports the pivotal role of the gut microbiome in the development of various diseases, including inflammatory bowel disease, obesity, and diabetes [37]. Recent studies have extended the investigation of microbiome biology to the ocular surface in oGVHD, shedding light on its potential involvement in the disease’s pathogenesis. In a study conducted by Li et al. [26], post-allo-HSCT patients exhibited diminished microbial diversity and a heightened presence of anelloviruses. OGVHD patients displayed distinctive microbiome profiles, with specific bacterial species such as *Gordonia bronchialis* and *Pseudomonas parafulva* showing correlations with the severity of oGVHD, thereby offering potential biomarkers. Further evidence revealed that while the ocular microbiome of healthy individuals was predominantly composed of a *Lactobacillus*/*Streptococcus* mix or *Corynebacterium*, *Corynebacterium* remained predominant in patients with dry eye disease and lax eyelid syndrome. Conversely, *Lactobacillus* predominated in oGVHD patients, who interestingly exhibited greater microbial diversity compared to their non-oGVHD counterparts [28]. Of note, they have a diminished microbial diversity compared to healthy controls. It is also worth noting that these results, while suggestive, did not reach statistical significance (*p* = 0.33). Similarly, Shimizu et al. [27] observed increased microbial diversity on the ocular surface of oGVHD patients compared to HSCT patients without GVHD. This intriguing finding hints at a potential association between different ocular surface diseases identified by the composition of their ocular surface microbiome.

Andersson et al. [23] explored the ocular surface microbiome of patients with aqueous tear-deficient dry eye, both with and without oGVHD. Their study revealed reduced microbial diversity and identified specific genera, such as *Pseudomonas* in healthy controls and Bacilli in patients with aqueous tear-deficient dry eye, as potential bacterial biomarkers. Interestingly, the identification of *Pseudomonas* as a potential biomarker for healthy controls contradicts the findings from Li et al. [26], which showed a correlation between *Pseudomonas parafulva* and the severity of oGVHD. Importantly, they also emphasized that their findings appeared to be primarily influenced by decreased tear secretion rather than the use of antibiotics, immunosuppressants, or the presence of Sjogren’s syndrome.

In summary, these investigations offer valuable insights into the involvement of the ocular surface microbiome in the pathogenesis of oGVHD and its potential as a source of biomarkers. However, the variability in the findings among the literature in this review highlights the current challenge in determining the consistency and reliability of this method as a diagnostic and monitoring tool for oGVHD. The differing methodologies employed in collecting and analyzing across different studies (see Table 2) may have contributed to this variability, suggesting the need for a standardized approach to collecting ocular microflora. Additionally, it is essential to recognize the continuous exposure of the ocular surface to the external environment, which can make the differentiation between the true microbiome and transient microbial presences complex. Furthermore, antibiotics and immunosuppressants, both locally and systemically, are commonly used in this patient population and could potentially alter the ocular surface microbiome, leading to erroneous interpretations of ocular health. Additional research is warranted to thoroughly evaluate the influence of these factors, alongside variables such as age, sex, and population, to comprehensively understand the role of the ocular surface microbiome in oGVHD.

### 4.7. Clinical Findings as Biomarkers

Unlike traditional biomarker research, which often focuses on molecular profiling or metabolic alterations, Pellegrini et al. [22] offer a quantitative and objective means of assessing corneal fluorescein staining, a hallmark of oGVHD. One notable aspect of this digital analysis tool is its ability to distinguish between oGVHD and other ocular surface diseases, like Sjögren’s syndrome. By revealing distinctive corneal fluorescein patterns characterized by higher circularity and roundness in oGVHD patients, the technique offers a potential tool for early and accurate disease identification. This differentiation could have profound implications in clinical practice, enabling timely referrals to specialists and tailored therapeutic interventions.

### 4.8. Limitations and Future Implications

While this review paper strives to mitigate biases, several inherent limitations are acknowledged. Firstly, we must consider potential biases introduced during our review process. Our search strategy, while comprehensive, may still miss relevant studies, leading to selection bias. Moreover, our exclusion of studies in languages other than English may introduce language bias, omitting valuable research. Even within articles published in English, authors from different countries use slightly different terminologies (e.g., “ocular manifestations of chronic GVHD” as opposed to “ocular GVHD”). Additionally, there is always the potential for publication bias and/or reporting bias, which systematic reviews like this are particularly at risk of since we are surveying pre-existing literature in this field. Furthermore, the focus on peer-reviewed journal articles means that potentially important grey literature, such as conference abstracts, reports, or theses, are left out.

Lastly, the subjectivity in data extraction is a notable limitation. Different reviewers may extract and interpret data differently, adding variability to our analysis. While we strive for transparency and minimize these limitations through our rigorous process, it is important to acknowledge these potential sources of bias when interpreting our findings. In summary, while adhering to PRISMA guidelines and the Newcastle–Ottawa bias assessment provides a robust framework for evidence synthesis, these limitations must be kept in mind in our assessment of the research landscape.

A key objective in identifying a biomarker is to improve diagnostic guidelines and expand therapeutic options. As the number of allo-HSCT procedures increases, more individuals are at risk of developing complications such as oGVHD. Thus, it is imperative to better understand biomarkers for this disease to improve patient outcomes. A non-invasive approach, such as tear sampling, seems to be the most promising method to screen patients, pre- and post-transplant. Ideally, the biomarkers specific to oGVHD and disease severity should be prioritized, even if the assay will eventually include a combination of different biomolecules, such as cytokines, proteins, and leukocytes. Additionally, automated imaging modalities, which have the potential to measure oGVHD-specific pathophysiology objectively and accurately, should be integrated to monitor disease. These biomarkers may function similarly to biomarkers of other disease states. For instance, in systemic lupus erythematosus, antibodies to single-stranded DNA are a biomarker used for diagnostics due to its high specificity of 95% [38]. Cancer antigen 19-9 has also become a useful biomarker to aid in diagnosing and prognosing pancreatic cancer [39]. Our goal is that soon, the biomarkers described in this review may similarly serve patients. Early detection of associated biomarkers allows for early diagnosis and prevention of disease progression. Additionally, changes in biomarker levels may signify treatment efficacy and better inform clinicians when alternative therapies are necessary. Perhaps oGVHD will become a disease with novel treatments and these biomarkers will become surveillance tools. Altogether, oGVHD biomarkers will improve patient outcomes and our understanding of disease pathophysiology; thus, it is vital to conduct further studies to validate these biomarkers in a multi-modal approach with larger sample sizes.

## Figures and Tables

**Figure 1 biomolecules-14-00102-f001:**
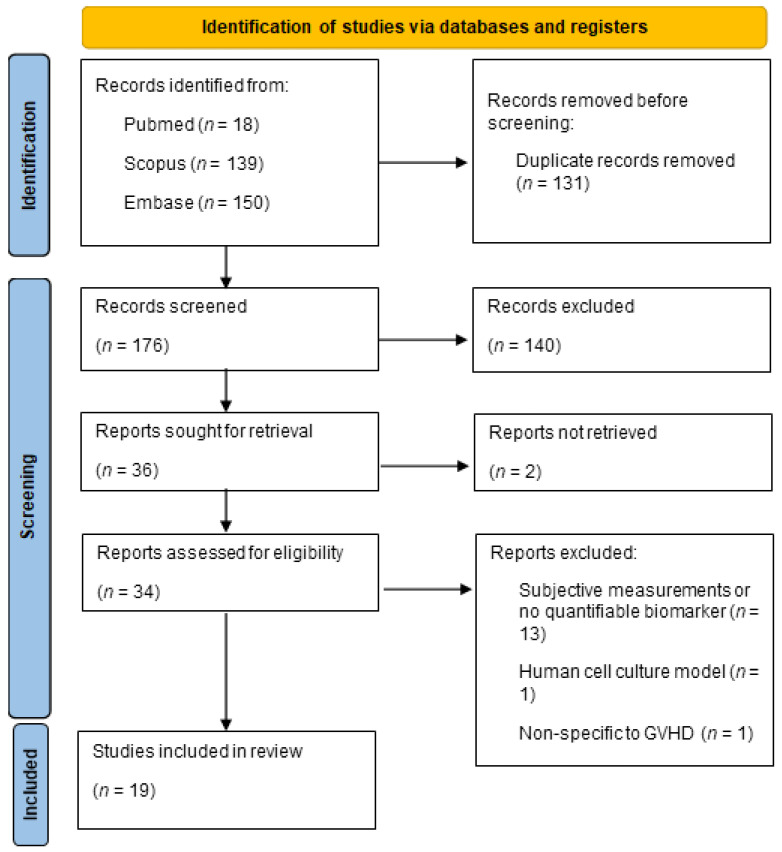
Flowchart depicting the paper count at each stage, from initial database identification to screening and final inclusion.

**Table 1 biomolecules-14-00102-t001:** OSDI, Shirmer’s score (without anesthesia), corneal staining, and conjunctival injection are given a score of 0–3. All four criteria are added for a maximum composite score of 11. The composite scoring for oGvHD severity is as follows: 0–4 (None), 5–8 (Mild/Moderate), and 9–11 (Severe) [3].

Score	0	1	2	3
OSDI Score	<13	13–22	23–32	≥33
Schirmer’s Test (in mm)	>15	11–15	6–10	≤5
Corneal Staining	No staining	Minimal staining	Mild/moderate staining	Severe staining
Conjunctival Injection	No injection	Mild/moderate injection	Severe injection	

**Table 2 biomolecules-14-00102-t002:** This table provides an overview of potential biomarkers associated with oGVHD, including details on sample types, patient groups, analysis techniques, comparisons with control groups, correlations with clinical parameters, and corresponding references.

Potential Biomarker(s)	Sample Type	Patient Groups	Technique	Comparison vs. Control	Correlations with Clinical Parameters	References
Cytokines: ICAM-1, IFN-β, IFN-γ, IL-1β, IL-1Ra, IL-2, IL-4, IL-5, IL-6, IL-7, IL-8, IL-10, IL-13, IL-12/23p40, IL-17A, IL-18, IL-21, IL33, LTA, TNFa, GM-CSF, BAFF, EGF	Tears	oGVHD, DED	Microsphere-based immunoassay analysis	oGVHD: ICAM-1, IL1β, IL-6, IL-8 higher than DED; IL-7 and EGF lower than DED.	Schirmer’s test: Neg correlations w/ ICAM-1, IL-6, IL-1β, and IL-8; pos correlations w/ EGF and IL-7. CFS: Pos correlations w/ ICAM-1, IL-6, IL-1β, and IL-8 (sig for IL-6 and IL-8). TBUT: Neg correlations w/ ICAM-1, IL-6, IL-1β, and IL-8 (sig for ICAM-1 and IL-8); pos correlations w/ EGF and IL-7. OSDI score: Pos correlations w/ ICAM-1, IL-6, IL-1β, and IL-8; neg correlations w/ EGF and IL-7. Elevated ICAM-1, IL-6, and IL-8 levels may signal disease severity.	Hu et al. [14]
Cytokines: IFN-γ, IL-10, MMP-9, IL-12, IL-13, IL17α, IL-1β, IL-2, IL-4, IL-6, IL-8, CXCL10, MCP-1, MIP-1α, RANTES, TNF-α; tear osmolarity	Tears and serum	oGVHD HSCT, healthy controls	Tearlab osmolarity system; serum sent to Allergen, Inc., Dublin, Ireland, for biomarker analysis	oGVHD had a higher median Oxford corneal staining score, lower RANTES, and higher IL-8 and TNF-α compared to non-oGVHD.	Elevated tear TNF-α positively correlated with higher corneal staining scores. Reduced tear RANTES levels associated with decreased Schirmer’s test and TBUT.	Pietraszkiewicz et al. [15]
Innate inflammatory cells, DNA products, cytokines, proteins	Ocular surface washing, peripheral blood (mice in vivo)	oGVHD HSCT, non-oGVHD HSCT, healthy controls	IHC, qPCR, Western blot	Elevated neutrophils and NET presence/activity correlates with oGVHD severity.	NET-related proteins, neutrophil elastase, MPO, OSM, NGAL, LIGHT/TNFSF14 cause corneal issues, conjunctival cicatrization, inflammation, and meibomian gland disease.	An et al. [16]
Proteins: TP, LYS-C, LACTO, LIPOC-1, TRANSF, ALB, ZAG-2 levels	Tears	oGVHD HSCT, pre-HSCT (same patients)	2100 BioAnalyzer (Agilent Technology, Santa Clara, CA, USA)	TP, LACTO, TRANSF, and ZAG-2 decreased in late-onset oGVHD patients.	No statistically significant differences were found in the clinical parameters between the oGVHD and pre-HSCT groups, and as such no correlational analyses between biomarkers and clinical parameters were made in this study.	Ciavarella et al. [17]
Cytokines: IL-6, IL-10, TNF-a, EGF, MMP-2, MMP-3, MMP-7, BAFF, and APRIL.	Tears and imaging (cornea)	oGVHD HSCT, non-oGVHD HSCT	Microsphere-based immunoassay analysis; confocal laser scanning microscopy	oGVHD: EGF, MMP-7, and APRIL significantly decreased compared to non-oGVHD HSCT.	Corneal epithelial scores significantly correlated with tear EGF and APRIL.	Liu et al. [18]
Protein: MMP-9	Tears	oGVHD, non-Sjogren DED	TearLab^TM^ Osmolarity System (TearLab^TM^ Corp., San Diego, CA, USA)	MMP-9 expression elevated in oGVHD patients.	Positive correlations between MMP-9 and conjunctival staining, OSDI, and Schirmer test.	Berchicci et al. [19]
T-Cells: CD8	Conjunctival impression cytology	oGVHD HSCT, non-oGVHD HSCT, healthy controls	Immunofluorescence	Significant difference in controls vs. pre- and post-HSCT w/ and w/o GVHD; no significant difference between oGVHD HSCT and non-oGVHD HSCT.	Association between CD8 T-cell levels and Schirmer’s test, TBUT, and OSDI.	Alba-Linero et al. [20]
Lipids: phosphatidylcholine, sphingomyelin, lactosylceramide, DHA	Tears	oGVHD HSCT, healthy controls	Mass spectroscopy	Phosphatidylcholine, sphingomyelin, and lactosylceramide correlate with clinical findings.	Phosphatidylcholine, sphingomyelin, and lactosylceramide: strong correlations with NIH eye score, TBUT, CFS, and Schirmer’s test.	Ma et al. [21]
Fluorescein uptake	Split lamp photography of corneal staining	Sjogren Syndrome, oGVHD HSCT	Corneal staining index via ImageJ 1.51s (National Institutes of Health, Bethesda, MD, USA)	Circularity and roundness of corneal staining index were higher in oGVHD vs. SS.	The corneal staining index was significantly correlated with Oxford and NEI scales.	Pellegrini et al. [22]
Microbiota	Swab samples from conjunctival fornix	Dry eye w/ and w/o GVHD, healthy controls	16S rRNA gene sequencing	Pseudomonas identified as a bacterial biomarker for controls and Bacilli for aqueous tear-deficient dry eye.	Three genera correlated with OSDI or Schirmer’s test: Staphylococcus in the OGVHD group and Chryseobacterium and Micrococcus in the dry eye group	Andersson et al. [23]
79 proteins (nucleic acid binding proteins, cytoskeletal proteins, transfer and receptor proteins, enzyme modulators, and hydrolases)	Tears	oGVHD HSCT, healthy controls	Liquid chromatography, mass spectrometry	54 out of 79 proteins were upregulated, encompassing nucleic acid binding and cytoskeletal proteins. Downregulated proteins included transfer and receptor proteins, enzyme modulators, and hydrolases. 36 newly identified proteins displayed altered expressions.	No direct correlations were made between clinical parameters and potential biomarkers. The study focused on following changes in protein levels between both groups. However, OSDI, TBUT, Oxford score, and Schirmers test are strongly associated with progression to ocular GVHD in this study.	Gerber-Hollbach et al. [13]
Proteins, untargeted proteomic methods	Tears	oGVHD HSCT, non-oGVHD HSCT	Liquid chromatography, mass spectrometry	Lactotransferrin, lysozyme, polymeric immunoglobulin receptor, immunoglobulin J chain, prolactin-inducible protein, and immunoglobulin heavy constant alpha downregulated with increased severity.	Reduced LYZ and PIGR linked to higher OSDI and Oxford staining scores. These proteins also had strong inverse relationships with overall NIH scores. Regarding PGAM1, weaker correlations were observed with clinical measures. PGAM1 levels were positively associated with Oxford score and OSDI but negatively correlated with the Schirmer test and TBUT scores.	O’Leary et al. [24]
Proteins: IL-6, IFN Gamma, IL-8, IL 10, IL-15, IL-4, IL-2, IL-12P70, IL-17A, VEGF, TNF-alpha, MMPs	Tears	oGVHD HSCT, non-oGVHD HSCT, healthy controls	Bio-Plex assay	oGVHD: elevated levels of interferon γ, IL-6, IL-8, IL-10, IL-12P70, IL-17A, MMP-9, and VEGF compared to controls. Tear MMP-7 and MMP-9 are higher in non-oGVHD, suggesting potential indicators for oGVHD in post-allo-HSCT individuals.	A negative correlation was found between Schirmer’s test values and IL-2, IL-4, IL-6, IL-12P70, MMP9, and VEGF. Tear proteins were significantly decreased in oGVHD eyes, while levels appeared similar in non-GVHD and control eyes. Notably, tear levels of MMP7 and MMP9 were significantly elevated in non-oGVHD patients compared to healthy controls.	Nair et al. [25]
Microbiota	Conjunctival sample	oGVHD HSCT, non-oGVHD HSCT, healthy controls	Metagenomic shotgun sequencing, sequencing libraries qPCR	Reduction in ocular surface microbiota diversity in both allo-HSCT w/ and w/o oGVHD, indicating microbial dysbiosis post-allo-HSCT, irrespective of oGVHD development.	*Paracoccus* sp. was negatively associated with corneal fluorescein staining score, while *Acidovorax* sp. had a positive association.	Li et al. [26]
Microbiota	Samples from the lower conjunctiva via a sterile cotton swab (without anesthesia)	oGVHD HSCT, non-oGVHD HSCT, healthy controls	Microbes were cultured	More species were detected in severe chronic ocular GVHD, implying diversified conjunctival microbial communities in GVHD patients.	Positive correlation between the number of detected species and the ICO score and an inverse correlation between the number of detected species and TBUT.	Shimizu et al. [27]
Microbiome	Samples from superior and inferior tarsal conjunctiva and inferior conjunctival fornixes (with anesthesia)	Chronic SJS, oGVHD HSCT, LES, DED, healthy controls	MiSeq (Illumina Inc. San Diego, CA, USA), Mothur software v. 1.31.2	Positive relationship between the number of detected species and the ICO score. These findings suggest that as the severity of chronic ocular GVHD grading increases, more bacteria can be detected.	Biopsy-confirmed SJS/TEN, ocular oGVHD per the Chronic Ocular Graft-vs-Host-Disease (GVHD) Consensus Group, LES as defined by van den Bosch, and DED diagnosed based on DEWS II criteria.	Zilliox et al. [28]
Lymphotoxin-α (LT-α)	Tears	oGVHD HSCT, healthy controls	LT-α concentration test strip, S03A analyzer	LT-α decreased significantly in oGVHD patients with no sex difference.	LT-α significantly correlated with OSDI, NIH eye score, T-BUT, and CFS among all participants.	Ma et al. [29]
Proteins: IL-2, IL-6, IL-8, IL-10, IL-12, IL-17, CCL2, GM-CSF, M-CSF, GMCSF, FGF1, FGF2, EGF, VEGF, Fas-L, BAFF, CD40L, CD137, APRIL, ICAM-1, CD62E CD106, MMP-2, MMP-3, MMP-7, neuropilin-1, PDGF-CC, NGF-beta, and TNF-alpha	Tears	oGVHD HSCT, dry eye disease	Microsphere-based immunoassay analysis	oGVHD: Elevated G-CSF, M-CSF, GM-CSF, FGF1, FGF2, BAFF, CD40L, CD137, CD106, CD62E, MMP2, MMP3, and neuropilin-1. Reduced APRIL.	IL-2, IL-6, IL-8, MMP-3, neuropilin-1, CD62E, CD52, and BAFF negatively correlated with TBUT. BAFF positively correlated with OSDI and CFS scores. CD62E and CD52 are positively associated with OSDI and NIH grades.	Shen et al. [30]
Compliment: C2, C3a, C3/C5a; Cytokines: IFN-b, IFN-g, IL-1b, IL-1Ra, IL-2, IL-4, IL-5, IL-6, IL-7, IL-8, IL-10, IL-13, IL-12, IL-17A, IL-18, IL-21, IL-33, TNF-a, GM-CSF, BAFF, EGF, ICAM-1, lipoteichoic acid	Tears	oGVHD HSCT, healthy controls	Microsphere-based immunoassay analysis	oGVHD: Differences were observed in C2, C3a, and C5/C5a. ICAM-1, IL-1b, IL-6, IL-8, BAFF, and GM-CSF exhibited increased levels compared to healthy controls. IL-7 and EGF were lower.	Schirmer: neg. correlated with ICAM-1, IL-6, IL-1b, IL-8, BAFF, GM-CSF, C2, and C3a, while pos. correlating with EGF, IL-7, and C5/C5a, significantly for IL-1b, EGF, GM-CSF, and IL-7. CFS: pos. correlations with IL-6, IL-1b, IL-8, BAFF, GM-CSF, C2, and C3a and neg. correlations with ICAM-1, EGF, IL-7, and C5/C5a. TBUT: neg. correlations with ICAM-1, IL-6, IL-1b, IL-8, BAFF, GM-CSF, C2, C3a, and C5/C5a, while pos. correlations with EGF and IL-7, particularly for IL-1b, GM-CSF, and C3a. OSDI: pos. correlations with IL-6, IL-1b, IL-8, BAFF, GM-CSF, C2, C3a, and C5/C5a and neg. correlations with ICAM-1, EGF, and IL-7, notably for IL-6, IL-1b, GM-CSF, and IL-7	Qiu et al. [31]

Abbreviations: ICAM, intercellular adhesion molecule; IFN, interferon; IL, interleukin; LTA, lipoteichoic acid; TNF, tumor necrosis factor; GM-CSF, granulocyte macrophage colony-stimulating factor; BAFF, B-cell activating factor; EGF, epidermal growth factor; MMP, matrix metalloproteinase; CXCL, chemokine ligand; MCP, Membrane cofactor protein; MIP, macrophage inflammatory protein; RANTES, regulated upon activation, normal T cell expressed and secreted; IHC, immunohistochemistry; qPCR, qualitative polymerase chain reaction; TP, total protein; LYC-C, lysozyme C; LACTO, lactoferrin; TRANSF, transferrin; ALB, albumin; ZAG-2, zinc-alpha-2-glycoprotein; APRIL, a proliferation-inducing ligand; DHA, docosahexaenoic acid; VEGF, vascular endothelial growth factor; CCL, chemokine liganc; FGF, fibroblast growth factor; PDGF-CC, platelet-derived growth factor-CC; NGF, nerve growth factor.

**Table 3 biomolecules-14-00102-t003:** This table displays the scores obtained through the Newcastle–Ottawa Quality Assessment. Individual scores for each category and the total score are presented.

	Selection				Comparability	Exposure			Total
	Is the Case Definition Adequate	Representativeness of the Cases	Selection of Controls	Definition of Controls	Comparability of Cases and Controls	Ascertainment of Exposure	Same Method of Ascertainment for Cases and Controls	Non-Response Rate	Total
Pietraszkiewicz, et al. [15]	☆	☆		☆		☆	☆	☆	6
Qiu et al. [31]	☆	☆	☆	☆		☆	☆		6
Shimizu et al. [27]	☆	☆		☆	☆	☆	☆		6
Li et al. [26]	☆	☆		☆	☆	☆	☆		6
O’Leary et al. [24]	☆	☆		☆	☆	☆	☆		6
Gerber-Hollbach et al. [13]	☆	☆		☆	☆	☆	☆		6
Liu et al. [18]	☆	☆		☆	☆	☆	☆		6
Alba-Linero et al. [20]	☆	☆		☆	☆	☆	☆		6
Hu et al. [14]	☆	☆		☆		☆	☆		5
Shen et al. [30]	☆	☆		☆		☆	☆		5
Ma et al. [29]	☆	☆		☆		☆	☆		5
Ciavarella et al. [17]	☆	☆			☆	☆	☆		5
Pellegrini et al. [22]	☆	☆		☆		☆	☆		5
Ma et al. [21]	☆	☆		☆		☆	☆		5
Andersson et al. [23]	☆	☆		☆		☆	☆		5
Zilliox et al. [28]		☆		☆		☆	☆		4
Nair et al. [25]		☆		☆		☆	☆		4
An, et al. [16]	☆	☆				☆	☆		4
Berchicci et al. [19]		☆		☆		☆	☆		4

☆ represents a criterion that the paper met during the assessment.

## Data Availability

No new data were created or analyzed in this study. Data sharing is not applicable to this article.

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
