# Peer review of "Molecular Biomarkers in Ocular Graft-versus-Host Disease: A Systematic Review"

_biomolecules, 2024, doi:10.3390/biom14010102_

Round 1

Reviewer 1 Report

Comments and Suggestions for Authors

The manuscript demonstrates the potential biomarkers for oGCHD diagnosis by summarizing 20 published references (2018-2023). Although it lacks a deep analysis or solid experimental data to validate the main conclusions, the summary of the new findings from the recent literature might give a hint to direct future studies.

The data presentation needs to be significantly improved:

1. Please remove the redundant words in Table 2, such as "suggest" " additionally" "and", etc. to make the table concise. 

2. Table 3 needs to be reorganized by switching the rows and columns. Please organize the broken table contents in an integrated table and rank the references by scores from 6 to 3.

3. A summary or a table is required to summarize the main identified biomarkers with the diagnostic standards to distinguish the oGCHD and other DEDs. For example, the increased levels of IL-6 and IL8 along with the reduction of IL-7 and EGF may indicate the oGCHD occurrence, in which how many folds of change (FOC) of these cytokines contribute to the diagnosis.

Author Response

Dear reviewer,

Thank you for your feedback on how to improve our manuscript. Here are the changes we have made:

  1. Tables 2 and 3 were reformatted and made more concise as suggested. Its organization and language now clearly summarize the biomarkers described by the different authors.

  1. During this review, we decided to exclude an additional article (Byun et al., 2019) due to the subjectivity of its biomarker measurements. We corrected this error by removing the article from the manuscript and editing the text/tables/figure accordingly since it did not meet the inclusion criteria.

  1. Lastly, since we believe that our revisions of the text and figures better summarize the findings of the articles included, we decided not to add an additional summary or table, as was suggested. 

Reviewer 2 Report

Comments and Suggestions for Authors

This review is poorly written. The following are a few examples:

1. The major table (table 2) is poorly made. Row 2, column 2, Sample type is "oGVHD, Non-oGVHD HSCT, healthy", which certainly is a careless mistake.

2. Table 2, Row 1-3 describe correlations with clinical parameters, however, row 4 and further down only listed the parameters without any comment on any correlation with anything. The rows on page 10 started describing correlations again... Obviously the table was hastily put together without even an internal standard.

Comments on the Quality of English Language

No comments

Author Response

Dear reviewer,

Thank you for your feedback on how to improve our manuscript. Here are the changes we have made:

  1. Tables 2 and 3 were reformatted and made more concise as suggested. Its organization and language now clearly summarize the biomarkers described by the different authors.

  1. During this review, we decided to exclude an additional article (Byun et al., 2019) due to the subjectivity of its biomarker measurements. We corrected this error by removing the article from the manuscript and editing the text/tables/figure accordingly since it did not meet the inclusion criteria.

Reviewer 3 Report

Comments and Suggestions for Authors

An excellent review of an important but complex issue. The advice for preoperative evaluation is important but requires multiple presentations by a variety of authors to be widely accepted. The ability to identify a valid biomarker may involve technology which does not rely upon tear collection as it is currently practiced given the multitufe of related factors uncovered. 

Author Response

(The authors gave the same response as above.)

Reviewer 4 Report

Comments and Suggestions for Authors

The paper entitled “Molecular Biomarkers in Ocular Graft-versus-Host Disease: A Systematic Review” is a study based on a systematic review of publications that focused on adult patients diagnosed with oGVHD following allo-HSCT, with a special interest in biomarkers.

The review showed that from the 20 articles included, 18 cytokine, proteomic, lipid, and leukocyte profiles were studied in the tear film, as well as ocular surface microbiota and fluorescein staining. The results showed that cytokine profiling is the most studied oGVHD biomarker in literature.

Considering that this is a systematic review, the authors should provide a PRISMA list and the number of PROSPERO registration of the Systematic Review. You can view the details with the two links: PRISMA list: http://www.prisma-statement.org/default.aspx?AspxAutoDetectCookieSupport=1; and PROSPERO registration: https://www.crd.york.ac.uk/prospero/

The study is of clinical interest and adds to current literature in this field, especially considering that biomarkers indicative of disease state may lead to a more targeted diagnosis and therapeutic approaches in these patients.

The paper is thorough and highlights the important issues behind this potential biomarker. The analysis and results are appropriate in supporting the findings. It has been correctly planned and represents a solid basis for future studies regarding potential markers for ophthalmic diseases. It is nicely written and of clinical interest. References are appropriate. The figures and tables are pertinent, and descriptive and assist in describing the results.

The authors should discuss how the results of these parameters could be useful in the management of patients and how this could change treatment options if this biomarker proves to be a good indicator of disease severity and progression. A flowchart of clinical assessments and treatments based on individual findings could add importance in a routine clinical setting.

Comments on the Quality of English Language

Minor editing and polishing can improve the flow of the paper.

Author Response

Dear reviewer,   Thank you for your feedback on how to improve our manuscript. Here are the changes we have made:   1. Tables 2 and 3 were reformatted and made more concise.   2. During this review, we decided to exclude an additional article (Byun et al., 2019) due to the subjectivity of its biomarker measurements. We corrected this error by removing the article from our manuscript and editing the text accordingly since it did not meet the inclusion criteria.    3. The PROSPERO application has also been completed and its registration information will be provided as soon as it is processed (this may take up to 10 days).   4. We added more in the final paragraph of the discussion section to discuss how the parameters could be useful in the management of this disease and how it could change treatment options if one or several of the biomarkers prove to be a good indicator of disease severity and progression. Furthermore, while acknowledging the potential value of a clinical assessment and treatment flowchart, we believe that this aspect was not within the scope of this systematic review. The primary focus of the paper is the identification of molecular biomarkers strongly correlating with disease state.

Round 2

Reviewer 1 Report

Comments and Suggestions for Authors

The table quality has been significantly improved.

Reviewer 3 Report

Comments and Suggestions for Authors
Corrections were accomplished in satisfactory manner

Reviewer 4 Report

Comments and Suggestions for Authors

The authors have addressed the issues satisfactorily.

Comments on the Quality of English Language

 Minor editing of English language required